# Automated Insulin Delivery (AID) Systems: Use and Efficacy in Children and Adults with Type 1 Diabetes and Other Forms of Diabetes in Europe in Early 2023

**DOI:** 10.3390/life13030783

**Published:** 2023-03-14

**Authors:** Marta Bassi, Daniele Franzone, Francesca Dufour, Marina Francesca Strati, Marta Scalas, Giacomo Tantari, Concetta Aloi, Alessandro Salina, Giuseppe d’Annunzio, Mohamad Maghnie, Nicola Minuto

**Affiliations:** 1IRCCS Istituto Giannina Gaslini, 16147 Genoa, Italy; 2Department of Neuroscience, Rehabilitation, Ophthalmology, Genetics, Maternal and Child Health, University of Genoa, 16126 Genoa, Italy; 3LABSIEM (Laboratory for the Study of Inborn Errors of Metabolism), Pediatric Clinic, IRCCS Istituto Giannina Gaslini, 16147 Genoa, Italy

**Keywords:** type 1 diabetes, insulin pumps, AID (automated insulin delivery), AHCL (advanced hybrid closed loop), APS (artificial pancreas systems), CSII (continuous subcutaneous insulin infusion), CGM (continuous glucose monitoring)

## Abstract

Type 1 diabetes (T1D) patients’ lifestyle and prognosis has remarkably changed over the years, especially after the introduction of insulin pumps, in particular advanced hybrid closed loop systems (AHCL). Emerging data in literature continuously confirm the improvement of glycemic control thanks to the technological evolution taking place in this disease. As stated in previous literature, T1D patients are seen to be more satisfied thanks to the use of these devices that ameliorate not only their health but their daily life routine as well. Limited findings regarding the use of new devices in different age groups and types of patients is their major limit. This review aims to highlight the main characteristics of each Automated Insulin Delivery (AID) system available for patients affected by Type 1 Diabetes Mellitus. Our main goal was to particularly focus on these systems’ efficacy and use in different age groups and populations (i.e., children, pregnant women). Recent studies are emerging that demonstrate their efficacy and safety in younger patients and other forms of diabetes.

## 1. Introduction

The management of Type 1 Diabetes (T1D) has changed substantially over the past years. Since the introduction of the first insulin pump into clinical practice which allowed continuous subcutaneous insulin infusion (CSII) in the 1970s, the scientific community and T1D patients’ goal for years has been the creation of an artificial pancreatic system (APS) [1,2,3]. APS is a closed-loop system capable of automatically administering an appropriate dose of insulin based on the level of blood glucose. The progress of technological evolution in the treatment of type 1 diabetes is represented in Figure 1.

Continuous glucose monitoring (CGM) allowed the development of the SAP (Sensor Augmented Pump), leading to clear improvement of glycemic control, although the SAP did not provide any interaction between sensor and insulin infusion [4]. Subsequently, SAPs were developed with the LGS (Low Glucose Suspend) function, capable of automatically interrupting the basal insulin infusion in case of hypoglycemia, and SAPs with the PLGS (Predictive Low Glucose Suspend) function, which consists of suspending basal insulin delivery in case of hypoglycemia and predicted hypoglycemia [5,6].

The first closed-loop systems studied on patients used a relatively simple algorithm, which worked only overnight (OCL—Overnight Closed Loop) [7]. Starting from 2015 to 2016, HCL (Hybrid Closed Loop) systems were finally introduced; these systems had algorithms capable of regulating insulin delivery based on the blood glucose during the night and day. These systems still require manual intervention by the patient for the decision on dosage and delivery of meal boluses, but they are able to regulate the delivery of basal insulin in complete autonomy, increasing or decreasing basal rate in relation to glycemia [8].

In the same years, dual hormone HCL systems capable of delivering both glucagon (or amylin) and insulin were extensively studied. Studies and trials performed with these instruments have not always shown superiority over single-hormone closed-loop systems and are not yet used in clinical practice [9].

Finally, in 2019 the AHCLs (Advanced Hybrid Closed Loop) were developed; AHCLs combine automated basal rate and correction boluses to keep glycemic values in a target range. Patients are only required to estimate carbohydrate consumption for meal boluses. These tools have proven to be more effective on glycemic control than HCLs. Early studies on the use of AHCL in adolescents and adults with type 1 diabetes have shown excellent results in terms of glycemic outcomes and patient satisfaction [10,11,12,13].

From 2013 onwards, many APS DIY (Do It Yourself) systems have been developed by healthcare professionals and patients, using algorithms that have not been approved yet by the scientific community but are available to patients through open-source platforms [14].

At the moment there is no device that can be considered totally closed loop (FCL—Full Closed Loop), since even the most advanced devices still require patient intervention for the administration of meal boluses. However, the recent significant technological advances of past years allow us to believe that the obstacles related to the management of blood sugar levels in the most difficult moments of daily life (meals, sports, intercurrent illnesses…) could be overcome soon by technological progress.

Systems which integrate insulin infusion with continuous glucose monitoring (CGM) are now widely used by T1D patients and evolving technologies offer the potential to highly improve glycemic control [15,16,17].

The purpose of this review is to present the characteristics of the AID systems currently available in Europe for T1D patients with a particular focus on their use and efficacy in relation to the different age groups. Some systems are approved only from a certain age, but many studies are emerging that demonstrate the effectiveness of using such tools even in very young children.

## 2. AID Systems

The characteristics of AID systems described in this section are summarized in Table 1. Pilot studies for each device are listed in Table 2. Randomized Control Trials (RCT) conducted on specific age groups are described in the third section of the review. Although CamAPS FX and Omnipod 5 do not fall within the definition of AHCL because they do not release automatic corrective boluses, we have chosen to include them in the review as they are able to vary the basal rate in a really important way and represent recent evolutions of previous systems.

### 2.1. Minimed 780G

Minimed 780G (Medtronic, Northridge, CA, USA) is the evolution of the first HCL pump introduced on the market in 2016 (Minimed 670G) and it has been approved for use from 7 years of age in Europe. The system is composed by an insulin pump, containing an interoperable Predictive Integrative Derivative (PID) algorithm, including adaptive insulin limits, which works paired with a dedicated sensor. Two types of sensors can be associated with Minimed 780G: Guardian Sensor 3, which requires capillary blood glucose calibration every 12 h, or the more recent Guardian Sensor 4, which does not require capillary calibration and must be replaced every 7 days. The insulin reservoir (available in two different capacities of 1.8 mL and 3 mL) and the infusion set must be replaced every 3 days for traditional kits and every 7 days for extended kits. The system is equipped with customizable alarms and alerts to allow safe use of the system (the most used alerts are those for hypoglycemia or hyperglycemia). Real-time glucose data sharing, remote glucose data viewing and automatic cloud upload via dedicated platforms (Carelink^®^, Medtronic, Northridge, CA, USA) are also available.

The pump functions in both manual or automatic mode. When working in manual mode, the pump administers insulin according to the basal schedules set and suspends delivery of basal insulin in case of predicted hypoglycemia. The manual mode is used for at least the first two days of use of the pump and is reactivated in case of loss of the sensor signal for more than 4 h, in case of delivery of maximum basal insulin according to the algorithm for more than 7 consecutive hours and in case of delivery of minimum basal insulin according to the algorithm for 3 to 6 h (depending on glycemic values).

The automatic mode (SmartGuard) uses an algorithm designed to maximize the patient’s time in euglycemia. The PID algorithm automatically calculates the basal insulin rates and the insulin sensitivity factor (ISF), regardless of the parameters set, based on the insulin needs of the previous 5–7 days. Insulin delivery is then adapted based on blood sugar values detected by the sensor [18]. The automatic mode works for daily insulin dosages above approximately 8 U/day; for lower dosages the instrument can be used in manual mode. As for the other HCLs, at the time of the meal patients must enter the amount of carbohydrates (CHO) they expect to consume and the pump suggests a bolus to be administered, based on the ICR (Insulin Carbohydrate Ratio) set and the blood glucose levels at the time of the bolus. The patient can choose to change or confirm the bolus suggested by the pump.

An additional feature, common to other AHCL systems is the administration of automatic correction boluses when the increase in basal insulin alone is not sufficient to bring blood glucose back to the desired target levels. Automatic boluses, if needed, can be delivered every five minutes.

#### 2.1.1. Specific Features

Specific features of MiniMed 780G, that differentiate it from other AHCL systems are the customizable blood glucose target at 100, 110 or 120 mg/dL and active insulin time (AIT) customizable from 2 to 8 h [19]. In addition, it is possible to activate a temporary target function at 150 mg/dL, which can be useful in situations where a smaller amount of insulin is needed, such as during physical activity; no automatic correction boluses are delivered in this mode.

#### 2.1.2. Pivotal and Approval Studies

MiniMed 780G pivotal trial was a single-arm, multicenter in-home trial of 157 individuals with T1D aged 14–75 years old (39 adolescents and 118 adults) who had been on pump therapy for at least 6 months. After a 2-week run-in period, participants were randomized to cross over from a target of 120 mg/dL to 100 mg/dL or 100 mg/dL to 120 mg/dL for 45 days. Time in Range increased from 68.8% at baseline to 74.5% at end of study and was higher for patients using a target of 100 mg/dL versus 120 mg/dL. Time Below Range decreased from 2.5% at baseline to 1.8% at end of study. Glycated hemoglobin (HbA1c) decreased from 7.5% at baseline to 7% at end of study and the percentage of study participants achieving the target HbA1c of <7% increased from 34% at baseline to 61% at end of study [12]. A dual-center study with 59 children, adolescents, and adults, compared AHCL to sensor-augmented pump therapy with predictive low glucose suspend (SAP + PLGS). TIR improved more in AHCL users than in the control group (70.4 ± 8.1% vs. 57.9 ± 11.7, *p* < 0.001), with greater improvement overnight and in the adolescents’ group [20]. Data from these pivotal trials of Minimed 780G in adolescents and adults with type 1 diabetes were encouraging in terms of glycemic outcomes and patient satisfaction.

#### 2.1.3. Real-World Studies

Subsequent real-world studies confirmed the greater effectiveness of Minimed 780G compared to HCL (Minimed 670G), PLGS (Minimed 640G) and multi-injection therapy in patients with T1D [21,22,23]. The use of the Minimed 780G system led to a reduction of time above range (TAR > 180 mg/dL) without increasing time below range (TBR < 70 mg/dL) in 52 patients (aged 15–65 years), that were well-controlled and experienced Minimed 640 users [21]. These findings are supported by other evidence that demonstrates safety and effectiveness in reducing hyperglycemia without increasing hypoglycemia in adolescents and young adults with type 1 diabetes using Minimed 780G compared to the previous Medtronic system with the HCL algorithm (Minimed 670G) [11]. A large retrospective study involving more than 4000 Minimed 780G users demonstrated that they achieved a mean GMI of 6.8% and a mean TIR of 76.2% after initiating AHCL, while minimizing hypoglycemia [22]. The real-world use of Minimed 780G also provides an increased level of patient satisfaction [23,24].

### 2.2. Tandem Control-IQ

Tandem Control-IQ Technology (Tandem Diabetes Care, San Diego, CA, USA) is an Advanced Hybrid Closed-Loop system approved for use from 6 years of age in both the USA and Europe. The system is composed of a t:slim X2 insulin pump, containing a Model Predictive Control (MPC) algorithm, which works paired with a Dexcom G6 (Dexcom, San Diego, CA, USA) sensor. The insulin reservoir (capacity of 3 mL) and the infusion set must be replaced every 3 days. The system is equipped with customizable alarms and alerts to allow safe use of the system. Real-time glucose data sharing, remote glucose data viewing and automatic cloud upload via dedicated platforms (Glooko^®^, Glooko Inc., Mountain View, CA, USA) are also available. In the USA, the t:connect app now allows automatic data upload into the cloud and allows users to view pump data and deliver boluses from their smartphone.

The automatic mode (Control-IQ) can be started immediately from the moment the pump is placed, setting the required parameters (weight, total insulin daily dose, ISF, ICR and basal rate). The glucose target is set between 112.5 and 160 mg/dL. The MPC algorithm is able to modulate basal insulin delivery based on 30-min glucose prediction, ISF and IOB (Insulin On Board). If sensor glucose values are predicted to drop or rise, the algorithm can automatically reduce or increase basal insulin infusion to totally suspend in case of predicted hypoglycemia (<70 mg/dL) or to release correction boluses in case of hyperglycemia (>180 mg/dL). The system can deliver a maximum of one correction bolus every hour.

#### 2.2.1. Specific Features

Control-IQ technology has two special modes to optimize glycemic control during the night and during sports. The Sleep Activity Mode works on a target range of 112.5–120 mg/dL; when the predicted glucose value is >120 mg/dL the pump increases the delivery of basal insulin, but it does not deliver correction boluses [25,26]. The Exercise Mode works on a target range of 140–160 mg/dL, allowing a reduced risk of hypoglycemia; when the predicted glucose value is <80 mg/dL the pump suspends the delivery of basal insulin.

#### 2.2.2. Pivotal and Approval Studies

There are several studies demonstrating greater effectiveness and safety of the Tandem Control-IQ system compared to previous therapy. In the pilot randomized controlled trial, the Control-IQ system was compared to sensor-augmented pump therapy (SAP) over 6 months on a cohort of 168 people with type 1 diabetes aged ≥14 years. Time in range increased by 10 percent from baseline with the closed-loop system (61% to 71%) and time spent in hypoglycemia was reduced compared to the control group. An improvement in HbA1c was also observed. After this study the Tandem Control-IQ system received FDA approval [27]. The following year, a similar study protocol was applied to 101 younger children with type 1 diabetes (aged 6–13). The Control-IQ system was compared with a sensor-augmented pump over 16 weeks. The closed-loop group’s percentage of time in target range was 11% higher than the control group, while the time spent in hypoglycemia was low in both groups. This study allowed Tandem Control IQ use in children aged 6 years and older [28].

#### 2.2.3. Real-World Studies

A 1 year retrospective study was conducted on 9451 child and adult patients previously on insulin therapy with the Basal-IQ system (PLGS SAP) that switched to the Control-IQ system (AHCL). Data showed an improvement in TIR by 10% using the Control-IQ system (73.6% vs. 63.6%, *p* < 0.001), with median percent time <70 mg/dL keeping as low as around 1% [29]. The use of the Control-IQ system also had important psychological effects on patients with T1D as demonstrated in a recent retrospective study: the 1435 participants who took part in the study by completing a questionnaire reported a marked improvement in quality of life and a reduced impact of diabetes on daily life. There was also an improvement in glycemic control with a time in range of 78.2% after the first 3 weeks of use of the new modality which further increased to 79.2% over the following 4 weeks [30].

### 2.3. CamAPS FX

CamAPS FX is a hybrid closed loop system developed by the CamDiab group residing at Cambridge University. CamAPS FX received the CE mark in 2020 and, at present, is the only closed-loop system approved for T1D from 1 year of age and during pregnancy. It consists of an inter-operable hybrid closed loop smartphone app containing an MPC adaptive algorithm. The algorithm works paired with Dexcom G6 CGM. The insulin pumps currently available to link with the app are Dana RS and Dana-I (Sooil Development, Seoul, Republic of Korea) and YpsoPump (Ypsomed Holding, Burgdorf, Switzerland). The insulin reservoir (YpsoPump capacity of 1.6 mL and Dana capacity of 3 mL) and the infusion set must be replaced every 3 days. The system is equipped with customizable alarms and alerts to allow safe use of the system. Glucose and insulin data are automatically uploaded on a cloud and remote monitoring is possible using the Diasend app developed by Glooko (Mountain View, CA, USA).

When CamAPS FX is started and connected to a sensor and pump, the algorithm constantly learns and adapts quickly to changing insulin needs. At baseline, the user has to set body weight and total insulin daily dose (TDD); the algorithm works in automatic mode for TDD superior to 5 UI/day. Basal insulin is administered every 8 to 12 min based on CGM glycemic values, regardless of the pre-programmed basal rate. The MPC algorithm automatically integrates information about total daily insulin requirements and post-prandial insulin requirements. The glucose target can be managed into up to 48 time-blocks during the day from 80 mg/dL to 200 mg/dL (default value 105 mg/dL) with 1 mg/dL adjustments. Boluses can be delivered directly from the smartphone app privately.

Although CamAPS FX does not fall within the definition of AHCL because it does not release automatic corrective boluses, we have chosen to include it in the review as it is able to vary the basal rate in a really important way, simulating the action of a corrective bolus.

#### 2.3.1. Specific Features

The main difference between CamAPS FX and the other AHCL systems is that the algorithm is contained in a mobile app, and it can potentially be applied to every insulin pump or CGM. Nowadays the app is only available for Android smartphones. The AIT (Active Insulin Time) is not customizable and is automatically and dynamically decided by the system.

CamAPS FX integrates two optional functions to automatically modify glucose target and insulin administration. “Ease-off” mode increases the glucose target and reduces the insulin delivery when glucose tends to decrease during physical activity; “Boost” mode reduces the glucose target and increases insulin delivery with automatic interruption when glucose values are below the target: this feature can be used in case of an increased insulin need at time of inactivity, increased food intake or during illness or stress.

#### 2.3.2. Pivotal and Approval Studies

The algorithm developed by the CamDiab group was at first integrated into an overnight closed loop system. The first small clinical trials comparing closed loop insulin delivery with CSII showed an increase in Time in Range and a reduced risk of hypoglycemia in the OCL group [7]. From 2013, the CamDiab group developed OCL into an HCL system, to both control basal insulin delivery during night and day. The first RCT was conducted on a small cohort of 12 adolescent inpatients comparing the new system with conventional pump therapy for 36 h. In the HCL group a significative increase of TIR was observed (84% vs. 49% for standard therapy), without any differences in TBR [31]. The following year a home-based study was performed and demonstrated the feasibility of unsupervised closed-loop insulin delivery under free-living conditions. A cohort of 17 adult patients was enrolled to compare the HCL system and SAP for 8 days with a significant increase in the time in range of the first group (75% vs. 62%) [32].

The pilot study with a large number of patients was conducted in 2018, comparing CamAPS FX to SAP in 86 T1D patients with suboptimal glycemic control in a 12-week observation period; an improvement of 11% of TIR and 0.4% of glycated hemoglobin was observed in the HCL group and glycemic outcomes were better in the HCL group [33].

#### 2.3.3. Real-World Studies

The first real world studies were conducted on the OCL system to test safety and efficacy in a home-based context. In 2014 two home-based studies were conducted among adolescents (for 3 weeks) and adults (for 4 weeks) to compare the OCL system with CSII. In both studies, the overnight time in range was significantly higher in the OCL group with a medium increase of 15% and 13.5%, respectively [34,35].

A similar study protocol was applied, the following year, on a population of 12 adolescents with suboptimal glycemic control (HbA_1C_ > 7.5 mmol/L) for 3 weeks. During the intervention period, TIR increased by a median value of 18.8%, without differences in TBR [36]. An increment of a median value of 10% in TIR was also observed in a cohort of 29 adults with well-controlled diabetes (HbA_1C_ < 7.5 mmol/L) for a period of 4 weeks [37]. Other key RCTs with patients in specific age groups have been conducted and are discussed in the section on age group studies.

### 2.4. DiabeLoop

The Diabeloop hybrid closed loop system was developed by the Diabeloop group of Grenoble, France. After receiving the Conformité Européenne (CE) mark in 2018, the system has become commercially available in Europe for the treatment of diabetes only in adults. It consists of a Model Predictive Control interoperable algorithm called DBLG1 contained in a locked handset. Two models of insulin pumps work with this system: Accucheck Insight insulin pump (Roche Diabetes CARE, Basel, Switzerland) and Kaleido insulin patch-pump (ViCentra, Utrecht, The Netherlands). The aforementioned pumps work paired with Dexcom G6 CGM. The insulin reservoir (Accuchek Insight capacity of 3 mL and Kaleido capacity of 2 mL) and the infusion set must be replaced every 3 days. Real time access to pump and sensor data is provided by a web platform (Diasend^®^).

The algorithm regulates the insulin administration every five minutes, regardless of the pre-programmed basal rates, based on the CGM glucose values. In addition, a correction bolus is administered if blood glucose is >180 mg/dL (this feature cannot be modified irrespective of hyperglycaemia threshold). The characteristic of this MPC algorithm is to predict the future insulin infusion rate based on collected data from previous insulin outputs and glucose values as inputs. In this way the loop can adapt to the single patient’s needs in terms of total daily insulin delivery and post-prandial and exercise-related insulin needs. Some specific settings can be modified by the patient such as the glucose target (default value 110 mg/dL, modifiable from 100 mg/dL to 130 mg/dL) and the hypoglycemia threshold for insulin suspension (default value 70 mg/dL, modifiable from 60 mg/dL to 85 mg/dL).

#### 2.4.1. Specific Features

DBLG1 automatically decides the way of administration of the bolus based on the starting blood glucose level, dividing the bolus calculated in a part administered straight away and a second part administered 30–90 min after the first one based on glycemic levels and type of meal (percentage of fat).

DBLG1 introduces the “Reactivity” feature that allows the patient to control the correction rate of a blood glucose value superior to the target. Reactivity in the hyperglycemic range regulates bolus insulin delivery (default value 100%, modifiable from 43% to 186%); reactivity in normoglycemic range regulates basal insulin delivery (default value 100%, modifiable from 59% to 147%); reactivity during meal regulates the rate of insulin delivered as meal bolus.

DBLG1 also introduces “Zen mode”: this feature reduces the risk of hypoglycemia in specific situations by raising the glucose target for a period from 1 h to 8 h (default value 20 mg/dL, modifiable from 10 mg/dL to 40 mg/dL). In addition, the activity mode allows an augmented temporary target (70 mg/dL), a reduction of the insulin delivered by the algorithm and an increase in insulin sensitivity for 14 h.

#### 2.4.2. Pivotal and Approval Studies

The first non RCT-pilot study, performed in 2014, proved the safety and effectiveness of the Diabeloop v1 algorithm for two five-hour post-prandial periods (with and without the algorithm) in a cohort of 12 adult patients. The primary outcome was the percentage of time spent in the target range of 70–180 mg/dL. With the limitation of low statistical power, TIR was higher during the test day (85.2%) than during the control day (69,2%). Percentage of TBR (<70 mg/dL) was 0.2% during the study day vs. 4.4% during the control day [38].

In 2018, a second pilot study, including eight adult patients previously on insulin pump therapy, a three-week in-home dual-center clinical trial showed that the Diabeloop closed loop system could provide positive metabolic outcomes. TIR was 70.2%, with 2.9% of TBR [39].

The first multicentric RCT was a 12-week home-based study conducted in 2019 on a cohort of 63 adult patients previously treated with an external insulin pump for at least 6 months. Patients received both treatment with the HCL algorithm and a sensor-augmented pump with a wash-out period of 8 weeks in between. Time in Range showed values significantly higher in the DBLG1 group (68.5%) than in the SAP group (59.4%). Although five episodes of severe hypoglycemia occurred in the DBLG1 group (vs. three in the SAP group), all associated to hardware disfunction or human error, time spent below the range (<70 mg/dL) showed significantly lower values in the DBLG1 group (2.0% vs. 4.3%) [40].

#### 2.4.3. Real-World Studies

Since the commercial availability, only one real-life study was performed evaluating data from 25 adults (already on insulin pump therapy) provided with a commercial DBLG1 system for 6 months. Glycemic control significantly improved with an increase in TIR from 53% to 69.7%. No serious adverse events and a significant decrease in time in hypoglycemia (from 2.4% to 1.3) were observed [41].

### 2.5. Omnipod 5

The Omnipod 5 HCL system developed by the Insulet group from (Acton, MA, USA), is the evolution of the commercially available Omnipod SAP system. This system is approved in the USA from 6 years of age and in Europe from 2 years of age. Like the previous version, the Omnipod 5 system is composed of a patch pump which works paired with Dexcom G6 CGM and the system can be managed with a mobile app or with a receiver. The insulin reservoir (capacity of 1.8 mL) and the patch pump must be replaced every 3 days. The system is equipped with customizable alarms and alerts to allow safe use of the system. Real-time glucose data sharing, remote glucose data viewing and automatic cloud upload via dedicated platforms (Glooko^®^) are also available. The Model Predictive Control (MPC) algorithm works on a glucose target which can be managed between 110 mg/dL and 150 mg/dL, with 10 mg/dL adjustments in eight time blocks during the day. When the closed loop system is activated, the insulin administration is regulated every 5 min, regardless of pre-programmed basal rates, based on CGM data, glycemic trend, and total daily insulin administration: the Smart-Adjust technology automatically updates the next pod with patient’s current insulin needs at every change.

#### 2.5.1. Specific Features

The distinctive feature is the MPC algorithm integrated into every individual patch pump. Different from other HCL systems when a bolus is delivered, the algorithm integrates the actual CGM value to calculate the bolus dose and takes account of the glucose trend to adjust the correction bolus amount. An “Activity feature” is also included which raises the glucose target to 150 mg/dL and automatically reduces insulin administration.

#### 2.5.2. Pivotal and Approval Studies

The first pivotal trial, including 111 children (aged 6 to 13) and 124 adults (aged 14 to 70), demonstrated safe and significantly improved HbA1c levels and time in target glucose range (+15.6% in children and + 9.3% in adults), with a very low occurrence of hypoglycemia (1.8% in children and 1.3% in adults). Only three severe hypoglycemic episodes occurred (one in children) and a DKA event in the children’s group [42].

No real-world data are available due to the recent commercialization.
life-13-00783-t001_Table 1Table 1Comparison of main technological characteristics of AHCL systems.
Minimed 780GTandem Control I-QCamAPS FXDiabeloopOmnipod 5**CGM*****No calibrations***Guardian 4*Duration: 7 days*
Dexcom G6*Duration: 10 days*
Dexcom G6*Duration: 10 days*
Dexcom G6*Duration: 10 days*
Dexcom G6*Duration: 10 days*
**Pump**MiniMed 780Gt:slim X2Dana RS—Dana I—YpsoPumpAccu-check Insight—KaleidoOmnipod**Algorithm**PID*On pump*MPC *On pump*MPC*App based*MPC*App based*MPC*On Pod***Reservoir capacity**1.8 mL3 mL (Extended kits)3 mL3 mL (Dana RS—Dana I)1.6 mL (YpsoPump)3 mL (Accu-check Insight) 2 mL (Kaleido)1.8 mL**Infusion set ****duration**3 days (Standard Kit)7 days (Extended Kit)3 days3 days3 days3 days**Glucose Target**100, 110 or 120 mg/dL*Adjustable*112.5–160 mg/dL*Non-Adjustable*105 mg/dL *Adjustable* between 80 and 200 mg/dL110 mg/dL *Adjustable* between 100 and 130 mg/dL*Adjustable* between 110 and 150 mg/dL**Active Insulin Time**2–8 h*Adjustable*5 h*Non-Adjustable*Adaptive learning*Automatically adjusted*Adaptive learning*Automatically adjusted*2–6 h*Adjustable***Adjustable settings**Glucose Target, AIT, ICRBasal rates, ICR, ISFGlucose Target, ICRGlucose Target, ICRGlucose Target, ICR**Adjustable features***Activity Mode*Target 150 mg/dL *Sleep mode*Target 112.5–120 mg/dL *Exercise mode*Target 140–160 mg/dL *Ease off mode* ↑ target*Boost mode* ↓ target*Zen mode*Raised target for a short period of time (1–8 h).*‘Reactivity’ feature**Activity Mode*Target 150 mg/dL **Correction boluses**Every 5 min*Automated*Every 1 h*Automated*No *(Incorporated into basal delivery)*Every 5 min*Automated*No *(Incorporated into basal delivery)***Phone-based bolusing**NoYes (USA only)YesYesYes**Remote Monitoring**Glucose dataGlucose data All data, SMS AlertGlucose data Glucose data **Data upload to Cloud***Automated**Automated* (USA only)*Automated*No*Automated*CGM—Continuous Glucose Monitoring, PID—Predictive Integrative Derivative, MPC—Model Predictive Control, AIT—Active Insulin Time, ICR—Insulin Carbohydrate Ratio, TDDI—Total Daily Insulin Dose. ↑ Raised ↓ Decreased.
life-13-00783-t002_Table 2Table 2Main pivotal, RCT and real-world studies for each HCL and AHCL system.
PopulationDurationComparatorSafety OutcomesGlycemic OutcomesReference**Minimed 780G***n* = 59*Aged 7–80 y*4 weeksSAP, PLGSNo SH eventsNo DKA events *(1 DKA in control)***TIR** ↑ 12% (57.9% to 70.4%)*No HbA1C data*Collyns 2021 [20] 
*n* = 157*Aged 14–75 y*3 monthsSAP, PLGS, HCLNo SH eventsNo DKA events**TIR** ↑ 6% (68.8% to 74.5%)**HbA1c** ↓ 0.5% (7.5% to 7.0%)Carlson 2022 [12]*Real world**n* = 4120*No data*2 months-*No data***TIR** ↑ 12% (63.4% to 75.5%)**GMI** ↓ 0.4% (7.2 % to 6.8%)Silva 2022 [22]
*n* = 113 *Aged 14–29 y*3 monthsHCL1 SH eventsNo DKA events**TIR** ↑ *(no mean difference data)***HbA1c** ↓ *(no mean difference data)*Bergenstal 2021 [11]**Tandem****Control-IQ***n* = 168*Aged 14–71 y*
6 monthsSAPNo SH events1 DKA event*(No DKA in control)***TIR** ↑ 11% (61% to 71%) **HbA1c** ↓ 0.3% (7.4% to 7.1%) Brown 2019 [27]
*n* = 101*Aged 6–14 y*4 monthsSAPNo SH eventsNo DKA events **TIR** ↑ 11% (53% to 67%)**HbA1c** ↓ 0.4% (7.6% to 7%) Breton 2020 [28]*Real world**n* = 9451*Aged 6–91 y*12 monthsPLGS*No data***TIR** ↑ 10% (63.6 % to 73.6 %)**GMI** ↓ 0.3% (7.2 % to 6.9%)Breton 2021 [29]**CamAPS FX***n* = 86*Aged 11–36 y*
3 monthsSAPNo SH events1 DKA event*(No DKA in control)***TIR** ↑ 11% (54% to 65%)**HbA1c** ↓ 0.4% (8% to 7.4%)Taushmann 2018 [33]
*n* = 31*Aged ≥ 18 y*1 monthSAPNo SH eventsNo DKA events **TIR difference** +10%(76.2% vs. 65.6%)*No HbA1C data*Bally 2017 [37] **Diabeloop***n* = 63*Aged ≥ 18 y*3 monthsSAP5 SH events *(3 SH in control)*No DKA events**TIR difference +9%**(68.5% vs. 59.4%)↓ **HbA1c difference −0.15%**(−0.29% vs. −0.14%)Benhamou 2019 [40]
*n* = 25*Aged ≥ 18 y*6 months-No SH eventsNo DKA events**TIR** ↑ 16% (53% to 69.7%)**HbA1c** ↓ 0.8% (7.9% to 7.1%)Amadou 2021 [41]**Omnipod 5***n* = 235*112 (6–13 y)**129 (14–70 y)*3 months-3 SH events1 DKA events**TIR** ↑ 15% 6–13 y(62.8% to 68.2%) **TIR** ↑ 9% 14–70 y(64.7% to 73.9%) **HbA1c** ↓ 0.71%(7.67% to 6.99%)Brown 2022 [42]SAP—Sensor Augmented Pump, PLGS—Predictive Low Glucose Suspend, HCL—Hybrid Closed Loop, SH—Sever Hypglycemia, DKA—Diabetic Ke-toacidosis, TIR—Time in Range, HbA1c—Glycated Hemoglobin.


## 3. Use of AHCL by Age Group

Table 3 shows the devices approved for use in T1D based on age groups and other circumstances. Table 4 lists the main RCT and real-world studies for each device, divided by age group.

### 3.1. Adult Age

All of the AID systems described are approved for adults. Most of the pilot studies and real-life studies of the AHCL that led to FDA (Food and Drug Administration) approval were conducted in adult or mixed adult and pediatric populations. However, there are some studies that have evaluated the effectiveness of these instruments only on the adult population. We report the most relevant studies on the largest number of patients currently available for each device.

Lepore et al. conducted a retrospective real-life study to evaluate the Minimed 780G system’s effect on glycemic control in adult patients with T1D. More than 100 patients with T1D, treated with four different insulin therapies (MDI, CSII, SAP with PLGS and HCL) were evaluated before, two months and six months after switching to the Minimed 780G system. Patients of all four groups achieved a mean GMI < 7%, TIR > 70%, TBR < 4%, and CV < 36%, which is recommended by the ADA Standard of Medical Care in Diabetes, including the MDI group with worse baseline glycemic control [43].

The efficacy of Tandem Control-IQ in adults was demonstrated also by Usoh et al. analyzing data from 66 patients who switched from multi-injection therapy to Tandem t:slim X2 from January 2020 to June 2021: it resulted in a significant increase in TIR (from 49.5% to 63.3%) and reduction in HbA1C levels (from 7.7% to 7.1%) [44].

CamAPS FX safety and efficacy was assessed even in older patients with a long duration of T1D, a population with risk factors for severe hypoglycemia. A multicentre RCT study was performed on a cohort of 37 patients aged > 60 years comparing CamAPS FX HCL system with SAP therapy for two 16-week periods. Hybrid closed-loop insulin delivery achieved superior glycemic control to SAP therapy (TIR 79.9% vs. 71.4%), without increasing the risk of hypoglycemia [45]. Another study investigated the application of Fiasp in the CamAPS FX HCL system in a cohort of 25 adults over an 8-week period of unrestricted living. Results showed that the use of Fiasp could provide a reduction in TBR compared with standard insulin aspart without compromising overall glycemic control: mean TIR value was 75% for both study groups thus showing excellent glycemic control in patients using CamAPS [46].

At present the DBLG1 system is only approved for use in adults. Due to the recent commercialization, at present only the overmentioned real life study has been performed [41].

However, some specific situations have been evaluated such as high carbohydrate intake meals or physical activity followed by uncontrolled carbohydrate intake were investigated in patients provided with the DBLG1 system. Data showed reduced glycemic excursions in patients treated with DBLG1 [47]. A post-hoc analysis RCT study confirmed no significant differences in time spent in hypoglycemia during days with and without physical activity [48].

### 3.2. From 6 to 18 Years Old

The AIDs approved for use in this age group are CamAPSx and Tandem Control-IQ. Minimed 780G can be used from 7 years old. Many pilot and real-world studies have been performed on a mixed adult and pediatric population, but some studies have evaluated the effectiveness of these tools only in the pediatric population. We report the most relevant studies on the largest number of patients currently available for each device which followed the pilot studies on the pediatric population.

A real-world study that evaluated the effectiveness of Minimed 780G on a large cohort of T1D patients aged 15 years or younger (*n* = 3211) showed that more than 75% of patients using the system achieved international consensus-recommended glycemic control with GMI of 6.8% and TIR of 73.9% [49]. A prospective study conducted on children and adolescents (aged 7–17 years) with T1D on MDI therapy who initiated the AHCL system following a 10-day structured protocol showed the achievement of the internationally recommended goals of glycemic control with TIR > 70% and a HbA1c of <7% in young patients previously treated with MDI [50]. Furthermore, a real-world study conducted by Tornese et al. to assess the effectiveness of advanced- (a-HCL) vs. standard-hybrid closed-loop (s-HCL) systems use after 6 months of treatment showed that children under the age of 14 years seem to benefit the most from a-HCL pumps as well as individuals with the worst glycemic control [51].

In a 6-month study involving 191 children (median age of 14 years) who started using Tandem Control-IQ, Laurel et al. have reported how the hybrid closed loop system improved TIR by 9% from baseline (57% to 66%, *p* < 0.001) [52].

A post hoc analysis of a multicenter study performed in 2022 including 46 patients aged 6–18, compared the CamAPS FX algorithm (N21) with SAP therapy (N25) over a 6-month period. The CamAPS FX subcohort achieved a TIR improvement of 15% and a significant decrease in HbA1_C_ mean levels (−1.05%) compared to the control group [53]. Due to the recent commercialization of the CamAPS FX closed loop app, only few real world-data are available. A single center perspective 3-month study including 39 children and adolescents, 9% of them using CamAPS FX, pointed out an improvement of glycemic control, diabetes management and quality of life for patients and caregivers [54]. The National Health Service pilot initiative in England which recruited 251 children and adolescents on insulin therapy with the commercially available HCL system (including CamAPS FX) showed similar results, including a reduction of hypoglycemia fear and a quality-of-life improvement for patients and caregivers [17].

Unless DBLG1 has not already been approved for treatment of diabetes in children and adolescents, in 2022 a randomized controlled non-inferiority trial was performed on a population of 17 patients aged 6 to 12 for two 6-week periods comparing the DBLG1 system with SAP therapy. The primary endpoint was to prove the safety of the DBLG1 system in a pediatric population in everyday life conditions: as in adult studies, TBR (<70 mg/dL) was significantly lower for the DBLG1 group (2.62%) than the SAP group (5.24%) and TIR was significantly higher in the DBLG1 group (66.2% vs. 58.7%) providing good metabolic control in prepubescent children with T1D and supporting the safe use of closed-loop technology in a pediatric population [55].

### 3.3. Children under 6 Years Old

The only algorithms currently approved for use under six years of age are CamAPS FX and Omnipod 5. However, given the widespread interest and expected benefit of using these devices even in younger children, many studies have started to demonstrate the efficacy and safety of other AHCL systems in this age group, some of which have been recently published.

In 2019, CamAPS FX HCL’s safety in toddlers and preschoolers with T1DM was assessed on a cohort of 24 children aged 1 to 7 years undergoing two 3-week periods under free-living conditions. Use of HCL with diluted insulin and standard strength insulin were compared without differences in TIR median values (72% vs. 70%) or TBR median values (4.5% vs. 4.7%). No severe hypoglycemia or diabetic ketoacidosis occurred in both cases [56]. An RCT, enrolling 74 children 1 to 7 years of age for two 6-week periods, comparing the Cambridge closed-loop system with SAP therapy, showed an improvement in TIR of 8.7% (71.6% vs. 62.9%), with a median time spent in the closed-loop mode of 95%, without an increase in time spent in hypoglycemia. The study further showed the feasibility of AHCL systems in this population [57].

A single arm study demonstrated the safety of the Omnipod 5 HCL system among a cohort of 80 young children aged 2 to 6 years. During the 13-week period no episodes of severe hypoglycemia or diabetic ketoacidosis were reported and there was an improvement in TIR of 10.9% [58].

The safety and impact of the Minimed 780G on glycemic control in 2- to 6-year-old children with type 1 diabetes were evaluated in a prospective study of 12 weeks. No events of diabetic ketoacidosis or severe hypoglycemia occurred. HbA1c, mean sensor glucose (MSG) value and TAR decreased, TIR increased significantly (+8.3%), whereas no significant changes in time below range (TBR) was observed. MiniMed 780G was demonstrated to be a safe and effective system in 2- to 6-year-old children with type 1 diabetes and wase associated with a reduction in parental diabetes distress [59].

Only one pilot study has been conducted and demonstrated the effectiveness and safety of the Control-IQ system in children between the ages of 2 and 5 years. This study involved 12 patients (range 2.0–5.8 years) diagnosed with T1D in at least the last 3 months. It was conducted for the first 48 h in a supervised hotel and subsequent 72 h at home under parental supervision. The results showed an increase in TIR during Control-IQ use compared to the baseline (from 61.7% to 71.3%). The percentage of time in hyperglycemia (>180 mg/dL) decreased from 34.1% to 25.7%. The percentage of time in hypoglycemia (<70 mg/dL) was similar on Control-IQ compared to baseline, while time < 60 mg/dL with Control-IQ was half the amount registered at baseline, but the difference was not significant [60].

Currently, there are no studies in the literature on the use of the DBLG1 system in children under the age of six. The evaluation of the efficacy and safety of all these devices constitutes a research field of primary interest in pediatric diabetology.
life-13-00783-t004_Table 4Table 4Main RCT or real-world studies on HCL and AHCL divided by age group.Adult Age





PopulationDurationComparatorGlycemic outcomesReference**Minimed 780G***n* = 1026 monthsMDI, CSII, PLGS, HCL**TIR** ↑ 26% (previous MDI)**TIR** ↑ 16% (previous CSII)**TIR** ↑ 10% (previous PLGS)**TIR** ↑ 9% (previous HCL)Lepore 2022 [43]**Tandem****Control-IQ***n* = 6618 months-**TIR** ↑ 14% (49.5 % to 63.3 %)**HbA1c** ↓ 0.6% (7.7% to 7.1%)Usoh 2022 [44]**CamAPS FX***n* = 374 monthsSAP**TIR** ↑ 9% (70% to 79.9%)**HbA1c** ↓ 0.8% (7.5% to 6.7%)Boughton 2022 [45]**Diabeloop***n* = 256 months-**TIR** ↑ 16% (53% to 69.7%)**HbA1c** ↓ 0.8% (7.9% to 7.1%)Amadou 2021 [41]**Children (6–18 y)**





**Population****Duration****Comparator****Glycemic outcomes****Reference****Minimed 780G***n* = 3211*Aged ≤ 15 y*6 months -**TIR** achieved 73.9%**GMI** achieved 6.8%Arrieta 2022 [49]
*n* = 34*Aged 7–17 y*3 months-**TIR** ↑ 36% (42.1% to 78.8%)**HbA1c** ↓ 2.1% (8.6% to 6.5%)Petrovski 2022 [50]
*n* = 44*Aged 2–21 y*6 monthsHCL↑ **TIR** difference +7%↓ **HbA1c** difference −0.5%Tornese 2021 [51]**Tandem****Control-IQ***n* = 101*Aged 6–14 y*4 monthsSAP**TIR** ↑ 11% (53% to 67%)**HbA1c** ↓ 0.4% (7.6% to 7%) Breton 2020 [28]
*n* = 191*Median 14 y*6 months-**TIR** ↑ 9% (57% to 66%)**HbA1c** ↓ 0.3% (7.5% to 7.2%)Messer 2021 [52]**CamAPS FX***n* = 39*Aged 2–18*3 months-**TIR** ↑ 16% (50.5% to 67%)**HbA1c** ↓ 0.5% (7.9% to 7.4%)Ng 2022 [54]**Diabeloop***n* = 17*Aged 6–12 y*6 weeksSAP**TBR** ↓ 3.3% (5.24% to 2.62%)**TIR** ↑ 7.5% (58.68% to 66.19%)Kariyawasam 2022 [55]**Children ≤ 6 y**





**Population****Duration****Comparator****Glycemic outcomes****Reference****Minimed 780G***n* = 353 months-TIR ↑ 8% (58.3% to 66.6%)↓ HbA1c—0.3%Pulkkinen 2022 [59]**Tandem****Control-IQ***n* = 125 days-TIR **↑** f7% (63.7 % to 71.3%)*No HbA1c data*Ekhlaspour 2021 [60]**CamAPS FX***n* = 744 monthsSAP↑ **TIR** difference +8.7%↓ **HbA1c** difference −0.4%Ware 2022 [57] ***Omnipod 5***n* = 803 months-**TIR** ↑ 10% (57.1% to 68.1%)**HbA1c** ↓ 0.5% (7.4% to 6.9%)Sherr 2022 [58]*** One serious adverse event of severe hypoglycemia occurred during the closed-loop period. No SH or DKA events in the other studies in children ≤ 6 y**SAP—Sensor Augmented Pump, PLGS—Predictive Low Glucose Suspend, HCL—Hybrid Closed Loop, SH—Sever Hypoglycemia, DKA—Diabetic Ketoacidosis, TIR—Time in Range, HbA1c—Glycated Hemoglobin, GMI—Glucose Management Indicator.


## 4. Use of AHCL in Other Forms of Diabetes or Special Populations

Table 5 shows the main studies conducted on the use of AID systems in other forms of diabetes or special populations.

### 4.1. Type 2 Diabetes

Approximately 6.3% of the worldwide population is affected by Type 2 Diabetes (T2D) with an increasing number of people on insulin therapy because of disease progression.

The earlier short-term inpatient studies were conducted by the CamDiab group. The first was performed in 2014 among a cohort of 12 insulin-naïve adult patients on glucose-lowering oral therapy. Results showed an increase in TIR from 24% at baseline to 40% after 24 h of treatment with HCL, without meal boluses [61]. During the following years more studies were conducted by the CamDiab group comparing insulin administration via AID with conventional subcutaneous insulin therapy. In 2018, Bailly et al. compared the use of HCL with conventional subcutaneous insulin among a population of 136 hospitalized patients with T2D for a maximum period of 15 days, with an increase of 24% in TIR (65.8% vs. 41.5%), a reduction of mean BG levels and glycemic variability [62]. Recently, the protocol was extended to a cohort of supervised outpatients with end-stage kidney disease on dialysis, provided with the CamAps FX algorithm, for a 20-day period, under free-living conditions. Data showed an improvement of 15.1% in TIR (52.8% vs. 37.7%), with a significant reduction of mean glucose values and time in hypoglycemia [63]. The most recent study has been published in 2023, demonstrating the safety and effectiveness of the CamAPS FX HCL system compared with standard insulin therapy in an open-label randomized crossover study including 26 adults with type 2 diabetes: TIR was 66.3% with closed-loop therapy versus 32.3% with control therapy. A significant reduction in HbA1c was also observed in the first group (7.3% vs. 8.7%) without any difference in TBR [64].

In 2022, a retrospective analysis among 500 Tandem Control-IQ patients with T2DM demonstrated improved glycemic control. A significant decline in mean glucose values from 166.8 mg/dL to 158.4 mg/dL was observed with a significant improvement in TIR from 64% to 72%, without any change in level 1 hypoglycemia and with a small, but statistically significant change in level 2 hypoglycemia [65].

The first outpatient feasibility study for AID in T2DM was performed in 2022 by Peters et al. comparing 8 weeks of Omnipod 5 system use with 2 weeks of standard therapy (MDI) among 24 patients with poorly controlled T2DM (HbA_1C_ > 8%). After 8 weeks of therapy, a 1.3% decrease of HbA_1C_ was observed, the mean sensor glucose levels decreased (mean difference, 29 mg/dL), TIR increased by an average of 19.2%, with a reduced time in hypoglycemia. The same cohort of patients was followed for an additional 13 weeks of AID use. During this period, the HbA1c level showed a modest further decrease (7.7% vs. 8.0%) without occurrence of severe hypoglycemia episodes [66].

Feasibility home-based studies highlight the potential of HCL and AHCL systems to improve glycemic control in patients with T2DM with suboptimal glycemic control on insulin therapy.

MiniMed 780G and Tandem Control IQ feasibility studies are still ongoing [67,68].

### 4.2. Cystic Fibrosis Related Diabetes (CFRD)

Cystic fibrosis-related diabetes (CFRD) is a common complication of Cystic Fibrosis that increases in incidence as patients age. Poor glycemic control has been shown to negatively impact lung function and weight, resulting in higher risk of recurrent pulmonary exacerbations [69]. To date we have little data on the use of AHCL systems in patients with CFRD. In a multicenter retrospective study continuous glucose monitoring of 13 patients with CFRD was compared before and after transition to Tandem Control-IQ. This study highlighted a significant increase in percentage of TIR, by 15.2% from baseline to 3 months (54.3% to 69.5% *p* = 0.001) as well as a decrease in percentage time in hyperglycemic ranges and glycemic variability. No significant variations regarding hypoglycemia were observed [70].

Further studies are needed to assess long-term efficacy in patients with CFRD as well as lung function and quality of life.

### 4.3. Type 1 Diabetes and Pregnancy

Women with pre-existing diabetes mellitus are at increased risk of pregnancy complications, such as congenital malformations, preeclampsia, and preterm delivery. Glycemic control is even more important during pregnancy for both mother and baby. For this reason, pregnant T1D patients have more restricted blood glucose targets than the general T1D population [71].

Out of the AID systems described, only CamAPS FX is authorized for use in pregnancy in women diagnosed with T1D. To date, there are no studies evaluating the efficacy and safety in pregnant women with T1D for Tandem Control-IQ and Diabeloop.

An RCT study involving 16 pregnant women with type 1 diabetes, compared overnight closed loop (OCL) use with SAP for two 4 week periods. OCL use was associated with an increase of time in the tighter glucose range recommended during pregnancy (63–140 mg/dL) compared with SAP therapy (74.7% vs. 59.5%; *p* = 0.002), without an increased risk of hypoglycemia [72]. Two years later the same group of studies involved 16 pregnant women using the closed loop system for 28 days and SAP therapy for as many days. The transition from one system to the other was separated by a washout period. Results showed that the time spent within the target (63–140 mg/dL) was similar in both cases, however pregnant women who used closed loop recorded significantly less time in hypoglycemia (0.24% vs. 0.47%) [73].

A study on six women with type 1 diabetes treated with SAP in one pregnancy and with Minimed 780G during the subsequent pregnancy was carried out in 2022. This study showed that women spent more time in the target glucose range (69.1% vs. 78.6, *p* = 0.045), less time above the target range and had lower glucose variability compared to their previous pregnancy when they used SAP [74]. AiDAPT (automated insulin delivery amongst pregnant women with type 1 diabetes) is a multicenter randomized controlled trial, currently still in progress, whose aim is to evaluate the impact of AHCL during pregnancy in women with type 1 diabetes. It is going to be the largest randomized study to date comparing automated closed loop and standard insulin delivery for pregnant women with T1D [75].

### 4.4. Patients with Uncontrolled Type 1 Diabetes (or Omitting Meal Boluses)

A particular population that deserves major attention in terms of technology use is that of patients with poorly controlled T1D. In fact, patients with elevated HbA1c values, poor adherence to self-monitoring of blood glucose and insulin therapy (frequent omission of meal boluses) were not considered candidates for HCL and AHCL therapy for a long time [76]. However, scientific evidence is demonstrating how much this group of patients could benefit from AHCL therapy.

Recently, Arktruk et al. conducted a real-life study to evaluate the effectiveness of the Tandem Control-IQ system in adults with T1D who omitted meal boluses. The analysis of the data of 30 enrolled patients showed a decrease in HbA1c of 1.6% and an increase in TIR of 19.3% without increasing TBR at one year follow-up. Therefore, Control-IQ proved to be effective and safe regardless of users’ engagement with the system [77].

The DBLG1 system showed its efficacy even in particular populations, such as patients with highly unstable diabetes: a cohort of eight patients on PLGS treatment with a poor TIR (median value 43.5%) and high glycemic excursions provided with the DBLHU (Diabeloop for highly unstable diabetes) system for a 4-week period improved their metabolic control (TIR median value 73.3%) [78].

Efficacy was also assessed in a population with excessive TBR: results of a post hoc analysis on a subset of 45 patients derived from three previous studies with time in hypoglycemia > 5% (median value of 7.9%) showed a decrease in TBR by more than 50% (median value 3.2%) with an improvement in TIR of 5% [79].

Considering the excellent results in terms of effectiveness of AHCL even in patients who omit meal boluses, some very recent studies have investigated the ability of these instruments to manage small amounts of carbohydrates without bolus, in order to determine the maximum amount of CHO which the system can automatically manage without incurring hyperglycemia after the meal. Tornese et al. demonstrated that unannounced snacks of up to 20 g of CHO can avoid hyperglycemia in Minimed 780G users [80]. Petrovski et al. have also demonstrated that the use of the Minimed 780G system with a preset of three personalized fixed carbohydrate amounts can lead to the achievement of the recommended glycemic targets; the simplified meal announcement may be a valuable alternative to CHO counting in users who are challenged by precise carbohydrate counting [81].
life-13-00783-t005_Table 5Table 5Main RCT or real-world studies on HCL and AHCL in particular populations or other forms of diabetes.Type 2 Diabetes






PopulationDurationComparatorSafety OutcomesGlycemic OutcomesReference**Tandem****Control-IQ***n* = 500*Aged ≥ 6 y*≥1 month-No SH eventsNo DKA events**TIR** ↑ 8% (64% to 72%)**HbA1c** ↓ 0.2% (7.3% to 7.1%) Forlenza 2022 [62]**CamAPS FX***n* = 20*Aged ≥ 18 y*3 weeksMDI1 SH eventsNo DKA events**TIR difference +15%**(52.8% vs. 37.7%)*No HbA1c data*Boughton 2021 [63]
*n* = 28*Aged ≥ 18 y*4 monthsMDINo SH eventsNo DKA events**TIR difference +35%**(66.3% vs. 32.3%)**HbA1c difference −1.4%**(7.3% vs. 8.7%)Daly 2023 [64]**Omnipod 5***n* = 22*Aged ≥ 18 y*21 weeks-No SH eventsNo DKA events*No TIR data ***HbA1c** ↓ 1.7% (9.4% to 7.7%)Davis 2022 [66]**CFRD**






**Population****Duration****Comparator****Safety outcomes****Glycemic outcomes****Reference****Tandem****Control-IQ***n* = 133 months-*Not indicated***TIR** ↑ 15% (54.3% to 69.5%)*No HbA1c data*
Scully 2022 [70]**Pregnancy**






**Population****Duration****Comparator****Safety outcomes****Glycemic outcomes****Reference****CamAPS ****(OCL)*****n* = 16****4 weeks****SAP**No SH eventsNo DKA events**TIR difference +15%**(74.7% vs. 59.5%)**Stewart 2016** [72]**Minimed 780G***n* = 61st pregnancy (SAP) 2nd pregnancy (AHCL)SAP*Not indicated***TIR difference +9%**(69.1% vs. 78.6%)**HbA1c difference −0.4%**(6.7% vs. 6.3%)Munda 2022 [74] **Uncontrolled T1D**






**Population****Duration****Comparator****Safety outcomes****Glycemic outcomes****Reference****Diabeloop***n* = 52 monthsPLGSNo SH eventsNo DKA events**TIR difference +29%**(43.5% vs. 73.3%)*No HbA1c data*Benhamou 2021 [78]
*n* = 45Post hoc analysis of three RCTs

**TIR** ↑ 5% (63.3% to 68.2%)Benhamou 2022 [79]


## 5. Comparison between AHCL Systems

In literature there is no strong evidence that better supports one AHCL system over another. Data collected from two groups of 90 patients, in a retrospective study, who switched to the AHCL system are compared (51 Minimed 780G and 39 Tandem Control-IQ). Both groups showed a remarkable improvement in glycemic control and reached the targets imposed by the International Consensus. Minimed 780G appeared to be superior in controlling hyperglycemia, whereas Tandem Control-IQ seemed to better prevent hypoglycemia [82].

In an observational, real-life, monocentric study, data of 31 children and adolescents who upgraded to AHCL were collected. An improvement in glucose control without, however, showing a significant difference between the two systems was confirmed [83]. A recent study confirms a significant improvement of glycemic control after one month of use and is then maintained at 1 year follow-up. This study highlights that Minimed 780G appears to be more effective in reaching the recommended glycemic targets. Further studies are necessary with larger population samples in order to concretize these results [84].

## 6. Future Perspectives in the Field of AID Systems

### 6.1. Ilet Bionic Pancreas

The iLet Bionic Pancreas hybrid closed-loop system developed by the Beta Bionics group (Boston, MA, USA), consists of an MPC adaptive algorithm embedded into an iLet insulin pump (Beta Bionics) that works paired with a Dexcom G6 CGM. It was first developed as a dual-hormone full closed loop system, then switched to a single-hormone approach. The glucose target can be managed from 110 mg/dL to 130 mg/dL in 10 mg/dL increments during the day.

#### 6.1.1. Specific Features

There are two main differences between the iLet system and the previous AHCLs. The MPC adaptive algorithm, only taking account of recent glucose data and insulin needs, automatically defines the basal insulin infusion rates and without any need of a pre-programmed one even when closed loop mode is inactive. In addition, meal boluses are inserted in a semiquantitative way in terms of size: the algorithm removes traditional user interactions making decisions for the patients.

#### 6.1.2. Pivotal Studies

In 2022, a pivotal RCT including 161 adults compared Beta Bionics artificial pancreas with patients’ standard of care for 13 weeks. An increment of 11% in TIR and a reduction of 0.5% in HbA1_c_ was observed in the first group without any significant differences in TBR between the two groups. Similar results were obtained among a cohort of 165 children (aged 6 to 17 years) undergoing the same protocol RCT study [85]. The iLet group showed an increase of 10% in TIR and a reduction of 0.6% in HbA1_c_. Safety was also assessed without any differences in TBR in the two study groups [86].

### 6.2. Do-It-Yourself System

Before the commercial availability of artificial pancreas systems, a community of T1DM patients and caregivers, the #WeAreNotWaiting movement, founded in 2013, began to support the use of do-it-yourself (DIY) artificial pancreas systems incorporating commercially available insulin pumps and continuous glucose monitors with automated open-source control algorithms [14].

A growing number of people with T1DM are choosing to use these systems due to the flexibility in terms of customization, such as individualized target glucose ranges, and the rapid development of algorithms [87].

The three main DIY closed-loop algorithms are OpenAPS, AndroidAPS, and Loop. No RCT data are currently available for any of the DIY systems. Even if extensive real-world data are available, healthcare providers report a lack of confidence in supporting, DIY artificial pancreases despite impressive observational and user self-reported improvements in glycemic variability, without any reported safety issues. This is probably due to the user-driven, unregulated nature of these systems [88].

Diabetes UK’s position statements about unapproved DIY closed-loop systems, released in May 2020, recommended that T1DM patients willing to use such systems should be aware that they do so at their own risk [89].

The non-profit company Tidepool (Palo Alto, CA, USA) sponsored by the Juvenile Diabetes Research Foundation and the Helmsley Charitable Trust, after acquiring the commercial rights, started developing a commercial version of the open-source Loop algorithm (Tidepool Loop) in 2020.

Real world data provided by 558 adults and children who used a community-developed Loop system for a 6-month period highlighted the safety and effectiveness of the DIY AHCL. TIR increased from a median value of 67% to 73%, with a reduction in the number of reported severe hypoglycemia events [90]. A randomized clinical trial using AndroidAPS and the DANA-I pump is currently ongoing [91].

In January 2023, after two years from submission, FDA approved the use of the Tidepool-Loop app: the software allows the interoperability between sensor and insulin pumps from different factories. We are still awaiting CE approval.

## 7. Conclusions

At the moment AID systems allow us to reach glycemic goals, something which was difficult with other types of treatment. A better glycemic control with the use of advanced therapies is redefining recommended glycemic targets in patients affected by diabetes that use technology [92]. With the development of several AID systems, some of which will be approved in the near future, the joint Diabetes Technology Working Group of the European Association for the Study of Diabetes and the American Diabetes Association has created a consensus statement providing a series of recommended targeted actions with a particular focus on AID systems’ safety [93].

The advantage of AHCL in terms of efficacy and patient satisfaction has been demonstrated in all age groups of T1D patients, even though further studies are necessary in order to approve such devices in younger children. It would also be interesting, from a clinical point of view, to explore their use in other types of insulin-dependent diabetes in future studies.

## Figures and Tables

**Figure 1 life-13-00783-f001:**
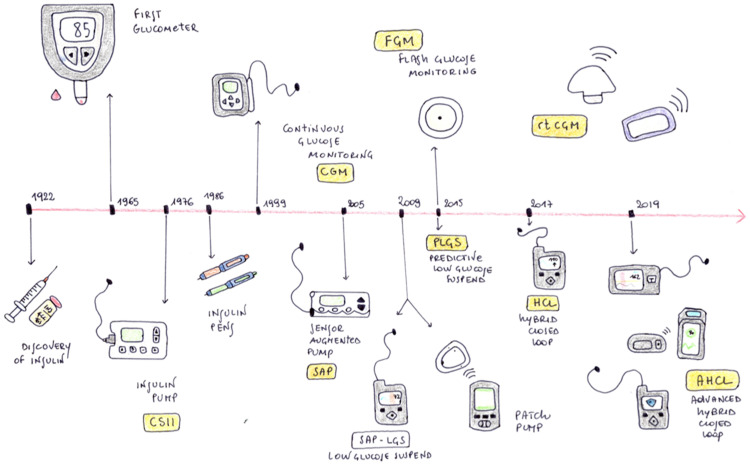
History of the evolution of technology for the treatment of type 1 diabetes.

**Table 3 life-13-00783-t003:** Advanced Hybrid Closed Loop (AHCL) systems and age groups.

	Minimed 780G	Tandem Control I-Q	CamAPS FX	Diabeloop
**Adults** (≥18 years)	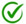	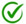	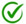	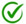
**Children** (≥6 years)	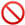	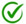	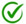	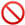
**Children** (≥7 years)	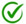	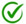	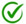	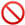
**Children** (1–6 years)	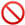	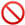	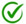	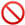
**Pregnancy in T1D**	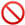	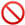	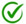	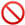

## Data Availability

Not applicable.

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
