# Peer review of "Automated Insulin Delivery (AID) Systems: Use and Efficacy in Children and Adults with Type 1 Diabetes and Other Forms of Diabetes in Europe in Early 2023"

_life, 2023, doi:10.3390/life13030783_

Round 1
Reviewer 1 Report
In this review, the authors summarized automated insulin delivery (AID). I think the content of this paper is very well organized and the Figure is also well drawn. This paper might be better if you also describe the contents of the consensus statement on Automated Insulin Delivery (Diabetes Care 45: 3058-3074, 2022.). Furthermore, it would be better to include a conclusion at the end of the text.
Author Response
We changed the manuscript as suggested
Reviewer 2 Report
The work presents a review that highlights the main characteristic of each advanced hybrid closed loop system available for patients with Type 1 Diabetes Mellitus and its efficacy and use in different groups.
Second page of the paper describes technological advance related to treatment in T1D, followed by a figure illustrating the timeline. The illustration is a freehand drawing, with ok images and handwriting, however, perhaps the use of a digital image for a timeline would make the manuscript look more serious, professional and adult.
Section 2. AHCL
“The characteristics of AHCL systems described in this paragraph …”
Perhaps paragraph should be replaced by SECTION
The characteristics of AHCL systems described in this SECTION …
RCT (line 90) has not been previously defined in the textclinical .
Table 1: Column Minimed 780G, row Algorithm: On pump) Extra bracket after pump
Regarding the reservoir capacity, text says that it is 1.8ml and 3ml, table only mentions 3ml.
Why reference [72] is only on text and it is not listed on Table 5?
The opposite happens to reference [11], which appears on Table 2, but do not appear on Section 2.1.2
Authors are invited to review these issues throughout the manuscript.
Author Response
- Regarding the Timeline Figure we have received different opinions from the reviewers, therefore we leave the decision on the possible removal to the editor.
- We changed the manuscript as suggested in the other comments
Reviewer 3 Report
Good work covering current clinical issues.
Suggests changing the name from AHCL system to AID system , in line with the current trend regarding the topic described.
I miss to add the number of units of insulin limiting the operation of various AID systems, e.g. Cam APS by 5 units; also missing is information on what details should be provided and possibly changed in various AID systems, e.g. in AHCL CamAPS we set the duration of insulin action, in CamAPS we need to update the patient's weight, etc.
Author Response
We changed the manuscript as suggested